# Detection of Coronaviruses and Genomic Characterization of Gammacoronaviruses from Overwintering Black-Headed Gulls (*Chroicocephalus ridibundus*) in Yunnan Province, China

**DOI:** 10.3390/microorganisms13040874

**Published:** 2025-04-10

**Authors:** Jun-Ying Zhao, Kan-Kan Chu, Pei-Yu Han, Ze Yang, Yi Tang, Wei Kong, Yun Long, Li-Dong Zong, Xing-Yi Ge, Yun-Zhi Zhang

**Affiliations:** 1Yunnan Key Laboratory of Screening and Research on Anti-Pathogenic Plant Resources from Western Yunnan, Key Laboratory for Cross-Border Control and Quarantine of Zoonoses in Universities of Yunnan Province, Institute of Preventive Medicine, School of Public Health, Dali University, Dali 671000, China; zhaojunying0714@163.com (J.-Y.Z.); ckk15709100745@163.com (K.-K.C.); hanpeiyu1511@gmail.com (P.-Y.H.); yangzeyz@163.com (Z.Y.); TangYiiii@outlook.com (Y.T.); kongwei4357@163.com (W.K.); 18080985193@163.com (Y.L.); zld2019106326@163.com (L.-D.Z.); 2Hunan Provincial Key Laboratory of Medical Virology, Institute of Pathogen Biology and Immunology, College of Biology, Hunan University, Changsha 410012, China

**Keywords:** *Gammacoronavirus*, black-headed gull, genome, phylogeny, cross-species transmission, genetic recombination

## Abstract

Black-headed gulls have been confirmed as the natural hosts of *Deltacoronavirus* (*δ-CoV*) and *Gammacoronavirus* (*γ-CoV*). A total of 59 CoV-PCR-positive fecal samples were identified among 509 fecal samples collected from overwintering black-headed gulls in Yunnan Province, China. The prevalence of black-headed gull deltacoronavirus (BHG-DCoV) was 3.54% (18/509), while that of black-headed gull gammacoronavirus (BHG-GCoV) was 8.06% (41/509). The prevalence of BHG-GCoV was significantly higher than that of BHG-DCoV (χ^2^ = 9.518, *p* < 0.01). Two complete genome sequences of BHG-GCoVs were obtained, with lengths of 27,358 bp and 27,355 bp, respectively, from the fecal samples of black-headed gulls. The nucleotide similarity between the two complete genomes is 98.75%. Phylogenetic analysis based on the whole genome has confirmed that the two strains of BHG-GCoVs clustered into the species *Gammacoronavirus anatis*. Although BHG-GCoVs belong to the species *Gammacoronavirus anatis*, they are distantly related to the representative strain Duck_CoV 2714 and exhibit a closer genetic relationship with GCoVs from *Xenus cinereus* (AvXc-GCoV) and *Numenius phaeopus* (AvNp-GCoV). Similarity analysis of the five conserved domains revealed a high amino acid similarity not only with AvXc-GCoV and AvNp-GCoV but also with GCoVs from common gulls detected in Poland and those from ruddy turnstones identified in Australia. Additionally, we found that, except for the common gull, the amino acid sequences of the S protein of BHG-GCoVs showed a 88.69% to 96.44% similarity with those of GCoVs carried by *Charadriiformes*, while the similarity with GCoVs carried by *Anseriformes* ranged from 31.15% to 54.81%. Furthermore, recombination events were detected in BHG-GCoVs, suggesting that these strains are likely recombinant strains of common gull GCoV and the GCoV of *Arenaria interpres* (AvAi-GCoV), indicating that recombination events may occur frequently among GCoVs.

## 1. Introduction

Since the 21st century, coronaviruses (CoVs) have caused three pandemics: severe acute respiratory syndrome (SARS) caused by SARS-CoV in 2002, Middle East respiratory syndrome (MERS) caused by MERS-CoV in 2012, and coronavirus disease 2019 (COVID-19) caused by SARS-CoV-2 in 2019. These pandemics have posed significant threats and resulted in considerable losses to the lives and economies worldwide. It is believed that these viruses originate from zoonotic viruses harbored in wild animals and infect humans through accidental cross-species transmission [1,2,3,4,5,6,7,8,9,10].

CoVs are positive-sense, single-stranded, non-segmented RNA viruses with typical spherical enveloped virions. Their membranes are anchored with glycoproteins (spikes) that encase a helical nucleocapsid, and the genome is approximately 27–32 kbp in length. The 2/3 region at the 5′ end of the genome encodes two large open reading frames (ORFs): ORF1a and ORF1b, which are translated into two large polyproteins (PPs), PP1a and PP1b. The remaining 1/3 of the genome encodes four major structural proteins: spike (S), envelope (E), membrane (M), and nucleocapsid (N). Several accessory proteins are interspersed among the structural proteins [11,12]. CoVs belong to the order *Nidovirales*, family *Coronaviridae*, and subfamily *Orthocoronavirinae* [13]. The International Committee on Taxonomy of Viruses (ICTV) divides the subfamily *Orthocoronavirinae* into four genera: *Alphacoronavirus* (*α-CoV*), *Betacoronavirus* (*β-CoV*), *Deltacoronavirus* (*δ-CoV*), and *Gammacoronavirus* (*γ-CoV*), based on the five conserved protein domains: 3C-like protease (3CLpro), RNA-dependent RNA polymerase (RdRp), Nidovirus RdRp-Associated Nucleotidyltransferase (NiRAN), zinc-binding domain (ZBD), and superfamily 1 helicase (HEL1). These domains create a profile that facilitates gross classification within the CoV taxonomy [1]. The genera *α-CoV* and *β-CoV* mainly infect mammals, while *δ-CoV* and *γ-CoV* mainly infect poultry and birds, with a small number also infecting mammals [14,15].

Infectious bronchitis virus (IBV), a representative CoV of the genus *Gammacoronavirus*, was first reported in the United States in 1931. IBV is highly contagious and primarily infects the respiratory system, kidneys, and reproductive system of poultry, leading to respiratory distress, kidney damage, and decreased egg production, which results in significant economic losses for the poultry industry. For more than 50 years following its discovery, IBV was the only known γ-CoV. However, over time, the virus and host diversity within the genus *Gammacoronavirus* has gradually been uncovered [16,17,18,19,20,21,22,23,24,25,26].

The black-headed gull (*Chroicocephalus ridibundus*), belonging to the family *Laridae* of the order *Charadriiformes*, is a migratory bird widely distributed across North America, Eurasia, northern, central, and southern Africa, the Indian subcontinent, the Indo–China Peninsula, the southwestern and southeastern coastal areas of China, as well as various islands in the Pacific Ocean [27]. In China, its primary breeding areas include Xinjiang, Inner Mongolia, and Heilongjiang. During winter, the black-headed gull migrates from Siberia to southern China, with Yunnan province serving as a typical wintering habitat. Black-headed gulls are hosts for several zoonotic viruses. As migratory birds capable of flying long distances, they are abundant, gregarious, and come into close contact with humans and other animals. These characteristics enable black-headed gulls to play a significant role in pathogen transmission, influencing the dynamics of various viruses and bacteria, as well as their ecology and evolution [14,20,28].

Previous studies have identified the black-headed gull as a natural host for *δ-CoV* and *γ-CoV* [29,30]. In our previous investigation of CoVs in black-headed gulls, we detected and analyzed the characteristics of three whole genomes (HNU4-1, HNU4-2, HNU4-3) of black-headed gull deltacoronaviruses (BHG-DCoVs) [31]. In this study, we collected fecal samples from black-headed gulls at three sampling sites in Kunming City, Yunnan Province, China, where we detected the presence of δ-CoV and γ-CoV using RT-PCR. For the positive samples of black-headed gull gammacoronaviruses (BHG-GCoVs), we performed whole-genome amplification of the two strains of BHG-GCoVs, which were subsequently bioinformatically analyzed to reveal their potential for cross-species transmission.

## 2. Materials and Methods

### 2.1. Sample Collection

In January 2023, a total of 509 fecal samples of black-headed gulls were collected from three sampling sites (Cuihu Park, Daguanlou Park, and Haigeng Dam) in Kunming City, Yunnan Province, China. All the samples were placed in cryopreservation tubes filled with 1 mL of virus transport medium (VTM). After collection, the samples were immediately put into the car refrigerator for storage and then transported to the laboratory and stored at −80 °C until use.

### 2.2. RNA Extraction and CoV Screening by RT-PCR

Approximately 70 μL of viral RNA was extracted from the fecal samples using the MagaBio plus Viral DNA/RNA Purification Kit III (Bioer, Hangzhou, China) through an automatic nucleic acid extractor (Bioer, Hangzhou, China) according to the manufacturer’s instructions. The extracted RNA was stored at −80 °C as a template for reverse transcription–polymerase chain reaction (RT-PCR). CoVs were initially screened by nested RT-PCR (Table 1). In the first round of CoV PCR amplification, a 602-bp fragment of the *RdRp* gene was amplified using the HiScript II One Step RT-PCR Kit (Vazyme, Nanjing, China). The 25 μL PCR mixture included 12.5 μL of 2 × One Step Mix (Dye Plus), 1.25 μL of One Step Enzyme Mix, 1 μL each of the forward and reverse primers, 6.25 μL of ddH_2_O, and 3 μL of the extracted RNA template. In the second round of CoV PCR amplification, a 440-bp fragment of the *RdRp* gene was amplified with 2 × Rapid Taq Master Mix (Vazyme, Nanjing, China), in which the 25 μL PCR mixture included 12.5 μL of 2 × Rapid Taq Master Mix, 1 μL each of the forward and reverse primers, 9.5 μL of ddH_2_O, and 1 μL of the product from the first round of PCR.

To confirm the host species, a 589-bp fragment of the *Cytb* gene was amplified using the 2 × Rapid Taq Master Mix (Vazyme, Nanjing, China) for host identification. The 25 μL PCR mixture included 12.5 μL of 2 × Rapid Taq Master Mix, 1 μL each of the forward and reverse primers, 9.5 μL of ddH_2_O, and 1 μL of the extracted DNA template (Table 1).

The PCR products were gel-purified (OMEGA Bio-tek, Norcross, GA, USA) and sent to Sangon Biotech for bi-directional sequence determination [32]. The sequences of the PCR products were compared with known sequences in the GenBank database.

**Table 1 microorganisms-13-00874-t001:** The PCR and primer information in this study.

PCR Format	Primer Name	Sequence (5′-3′)	PCR Reaction Protocol
The first round of CoV RT-PCR	AvCoV-F1 AvCoV-R1	GGKTGGGAYTAYCCKAARTG TGYTGTSWRCARAAYTCRTG [24]	The mixture was subjected to reverse transcription at 50 °C for 30 min, followed by pre-denaturation at 94 °C for 3 min, and then 35 cycles of denaturation at 94 °C for 30 s, annealing at 48.0 °C for 30 s, and extension at 72 °C for 36 s, with a final extension at 72 °C for 7 min.
The second round of CoV RT-PCR	AvCoV-F2 AvCoV-R2	GGTTGGGACTATCCTAAGTGTGA CCATCATCAGATAGAATCATCAT [24]	The mixture was subjected to pre-denaturation at 95 °C for 3 min, followed by 35 cycles of denaturation at 95 °C for 15 s, annealing at 48.5 °C for 15 s, and extension at 72 °C for 26 s, and finally an extension was performed at 72 °C for 5 min.
Host identification	L0 H0	GGACAAATATCATTCTGAGG GGGTGTTCTACTGGTTGGCTTCC [33]	The mixture was subjected to pre-denaturation at 95 °C for 3 min, followed by 35 cycles of denaturation at 95 °C for 15 s, annealing at 52 °C for 15 s, and extension at 72 °C for 36 s, and finally an extension was performed at 72 °C for 5 min.

### 2.3. Complete Genome Sequencing

Two complete genomes of BHG-GCoVs were amplified and sequenced using RNA extracted from black-headed gull feces as templates. RNA was amplified with degenerate primers, which were designed by multiple alignments of other coronavirus genomes with the complete genome, using Primescript One Step RT-PCR kit version 2 (Takara, Beijing, China). Additional primers were designed according to the results of the first and subsequent rounds of sequencing (Appendix A). The 5′ and 3′ genome end sequences were obtained by 5′ and 3′ RACE (Roche, Basel, Switzerland), respectively. The expected size of the PCR products was purified by gel and directly sequenced. The sequence was assembled to obtain the whole-genome sequence [31].

### 2.4. Genomic and Phylogenetic Analysis

The whole genome was annotated through Geneious Prime software (v2025.0.2). Multiple sequence alignments with other CoVs were conducted using MAFFT (v7.490) [34]. The most appropriate aa substitution model was calculated by the Find Best DNA/Protein Models tool in the MEGA X software (v10.1.8), and the model with the lowest Bayesian Information Criterion (BIC) score was taken as the best substitution model [35].

The phylogenetic tree was constructed based on the whole genomes of CoVs using the maximum likelihood method. The phylogenetic trees for other non-structural and structural protein sequences were constructed using the neighbor-joining method, with the p-distance adopted as the substitution model. The setting included “Substitution to include” set as d: Transitions + transversions, “Rates among Sites” set as uniform rates, and “Gaps Data Treatment” set as pairwise deletion. Bootstrap replicates were all set to 1000 times. The phylogenetic trees were visualized using the iTOL v6 online tool [36].

The similarity comparison of the genomes was carried out using BioAider v1.334 software [37]. The recombination events of BHG-GCoVs were detected using the RDP package v.4 and Simplot v3.5.1 [38].

### 2.5. Protein Tertiary Structure Analysis

The tertiary structure of the S protein was predicted and modeled using AlphaFold3 (https://alphafold3.org/, accessed on 10 January 2025), and the model results were visualized using ChimeraX v1.8 software.

### 2.6. Estimation of Divergence Dates

The *RdRp* gene was aligned using the MAFFT program v7.520 with the codon method in BioAider v1.334. Temporal structure in these *RdRp* gene sequences was analyzed using the TreeTime program [39]. The correlation coefficient (R^2^) was 0.01, indicating a very weak signal between sampling time and genetic distance. Therefore, a uniform prior distribution value (from 8 × 10^−5^ to 2 × 10^−4^ subs/site/year) was used, based on the latest report of the evolutionary rate of *RdRp* gene in DCoV [31]. A Markov chain of 10 million steps was run, with sampling conducted every 1,000 steps in BEAST v1.10.4 [40]. The mean evolutionary rate and the time of the most recent common ancestor (tMRCA) were calculated under an uncorrelated lognormal relaxed clock. The most appropriate nucleotide substitute model was GTR+F+G4, as calculated by ModelFinder according to the BIC method. Effective sample size (ESS) of parameters were checked in Tracer v1.7 program, ensuring that all reached convergence (ESS > 200). Finally, the maximum clade credibility (MCC) tree was obtained by discarding the first 10% of states in Tree Annotator package [41].

### 2.7. Statistical Analysis

Statistical analysis was performed using IBM SPSS Statistics 25 software. The chi-square (χ^2^) test was used to calculate the significant differences in prevalence.

## 3. Results

### 3.1. Detection of CoVs in the Feces of Black-Headed Gulls

A total of 59 CoV-PCR-positive fecal samples were detected from the 509 collected fecal samples of black-headed gulls at three sites (Cuihu Park, Daguanlou Park and Haigeng Dam), resulting in an overall prevalence of 11.59% (59/509, 95%CI: 8.80% to 14.38%). Among these, the prevalence of BHG-DCoVs was 3.54% (18/509, 95%CI: 1.90% to 5.10%), while the prevalence of BHG-GCoVs was 8.06% (41/509, 95%CI: 5.68% to 10.43%). The prevalence of BHG-GCoVs was significantly higher than that of BHG-DCoVs (χ^2^ = 9.518, *p* < 0.01). (Table 2).

A phylogenetic tree for the 59 detected BHG-CoVs was constructed based on partial RdRp sequencing results (γ-CoV about 393bp, δ-CoVs about 435bp), using the neighbor-joining method (Figure 1). The results showed that 41 strains of BHG-GCoVs (GenBank accession numbers: PQ676674, PQ676677–PQ676682, PQ676685–PQ676687, PQ676689, PQ676690, PQ676692, PQ676694, PQ676695, PQ676697–PQ676701, PQ676704, PQ676706, PQ676710–PQ676715, PQ676717–PQ676720, PQ676723–PQ676729, PQ676731, PQ676732) had a close genetic relationship with the ruddy turnstone CoV (MT993597) from Australia. The partial RdRp nucleotide similarity among all the detected BHG-GCoVs in this study ranged from 97.96% to 100%.

Additionally, BHG-DCoVs formed two subclades. Among these, 10 strains (GenBank accession numbers: PQ676675, PQ676676, PQ676683, PQ676684, PQ676688, PQ676691, PQ676693, PQ676702, PQ676707, PQ676716) formed a separate branch and appeared to be the ancestors of HKU27 (LC364342), HKU28 (LC364343), and HKU29 (LC364344). The other 8 strains (GenBank accession numbers: PQ676696, PQ676703, PQ676705, PQ676708, PQ676709, PQ676721, PQ676722, PQ676730) showed a close genetic relationship with HNU4-1 (OL311150), HNU4-2 (OL311151), and HNU4-3 (OL311152). The partial RdRp nucleotide similarity among all detected BHG-DCoVs in this study ranged from 93.79% to 100%. Host molecular identification was performed on all positive samples through *Cytb* gene amplification, confirming that the hosts were black-headed gulls.

### 3.2. Genome Structural Analysis of BHG-GCoVs

To further elucidate the genetic and evolutionary characteristics of the GCoVs detected in the feces of black-headed gulls, whole-genome amplification and sequencing were performed on two GCoVs-positive samples. This resulted in the acquisition of two whole-genome sequences of BHG-GCoVs with 27,358 bp and 27,355 bp, respectively. The nucleotide similarity between the two whole genomes was 98.75%, and they were designated as DLU1 (GenBank accession number PQ676672) and DLU2 (GenBank accession number PQ676673).

Genome annotation of the two strains of BHG-GCoVs, using AvXc-GCoV (PP845453) from the terek sandpiper (*Xenus cinereus*) in China as a reference, identified the genomic order of DLU1 and DLU2 as 5′UTR-ORF1ab-spike (S)—envelope, (E)—membrane, (M)—ORF5a-ORF5b-5a-5b-Nucleocapsid, and (N)—3′UTR. Comparisons with AvXc-GCoV, Shelduck_GCoV, and Duck_GCoV 2714 revealed that both DLU1 and DLU2 contained the typical genes of the *orthocoronavirus*. Additionally, all five strains contained the 5a and 5b accessory proteins. However, the BHG-GCoVs lacked the putative proteins ORF9a and ORF9b when compared to AvXc-GCoV, as well as the 3a and 3b accessory proteins in relation to Shelduck_GCoV and Duck_GCoV 2714 (Figure 2).

### 3.3. Genome Similarity Analysis of BHG-GCoVs

We performed BLAST (https://blast.ncbi.nlm.nih.gov/Blast.cgi, accessed on 10 September 2024) search on the GenBank database using the genomic sequences of BHG-GCoVs and selected 21 strains of GCoVs (Appendix A) that were significant for comparison with BHG-GCoVs (https://blast.ncbi.nlm.nih.gov/Blast.cgi. accessed on 19 December 2024). Among these, 13 strains were whole-genome sequences recently detected from 13 different hosts, representing three families across two orders, mainly the *Scolopacidae* and *Anatidae*, in China. Additionally, three GCoVs strains were detected in Australia and Poland, for which whole genomes or relatively lengthy genome sequences had been obtained. The remaining five strains were representative strains of the genus *Gammacoronavirus*.

The genomes of BHG-GCoVs show high nucleotide similarity to the GCoV (AvXc-GCoV) from *Xenus cinereus* and the GCoV (AvNp-GCoV) from whimbrels (*Numenius phaeopus*), with similarities ranging from 94.29% to 94.70%. In contrast, there is a notable difference in nucleotide similarity with Duck_GCoV 2714, the representative strain of *Gammacoronavirus anatis*, which is only 74.38% (Appendix A).

Five conserved replicase domains in non-structural proteins (3CLpro, NiRAN, RdRp, ZBD, and HEL1) are utilized for the classification of CoV [42]. To confirm the classification, we compared the nucleotide (nt) and amino acid (aa) similarities of these five domains between BHG-GCoVs and the selected 21 strains of GCoVs. The results indicated that the average aa similarity of these five domains in BHG-GCoVs ranged from 99.14% to 99.53% when compared to AvXc-GCoV, AvNp-GCoV, common gull GCoV, and ruddy turnstone GCoV. The aa similarity with Duck_GCoV 2714 was 90.75%. However, the aa similarity with the other four representative strains was only between 67.68% and 83.42% (Appendix A).

Among the structural proteins, the gene encoding the S protein is the most prone to mutations in CoVs. The S protein of BHG-GCoVs consists of 1,123 aa and has a unique S1/S2 cleavage site (520-FTPEKQVLDIKRFTSNYTEYAPMVFSD-546) [30], forming two subunits, S1 (546 aa) and S2 (577 aa). Similarity analysis of the S protein showed that the aa similarity between BHG-GCoVs and the GCoV of ruddy turnstone (*Arenaria interpres*) (AvAi-GCoV) ranges from 96.26% to 96.44%. In the S1 subunit, there are 37 aa differences between BHG-GCoVs and AvAi-GCoV, resulting in a similarity of 93.22%. In the S2 subunit, there are only 2 aa differences, yielding a similarity of 99.65%. The tertiary structure of the S protein of DLU1 and AvAi-GCoV is shown in Figure 3A. Interestingly, the aa similarity of the S protein of BHG-GCoVs varies from 88.69% to 96.44% with that of GCoVs from *Charadriiformes* (AvXc-GCoV, AvNp-GCoV, AvAi-GCoV, AvCt-GCoV, AvCa-GCoV, AvTt-GCoV, AvLl-GCoV, excluding the common gull GCoV). In contrast, the similarity with GCoVs from *Anseriformes* (AvAc-GCoV, AvAa-GCoV, AvAp-GCoV, AvAz-GCoV, AvMp-GCoV, AvMs-GCoV, Shelduck_CoV, Duck_GCoV 2714) is significantly lower, ranging from 31.15% to 54.81%. It is noteworthy that although both the common gull and the black-headed gull belong to the family *Laridae*, the aa similarity of the S protein of their GCoVs is only 54.51% to 54.59% (Appendix A).

In the S1 subunit, the aa similarity between BHG-GCoVs and GCoVs from *Charadriiformes* (AvXc-GCoV, AvNp-GCoV, AvAi-GCoV, AvCt-GCoV, AvCa-GCoV, AvTt-GCoV, AvLl-GCoV, excluding the common gull GCoV) ranges from 83.15% to 93.05%. In contrast, the aa similarity between BHG-GCoVs and GCoVs from *Anseriformes* (AvAc-GCoV, AvAa-GCoV, AvAp-GCoV, AvAz-GCoV, AvMp-GCoV, AvMs-GCoV, Shelduck_CoV, Duck_GCoV 2714) is only from 16.15% to 39.65%, while the similarity with the common gull GCoV is from 37.63% to 37.81%. In the S2 subunit, the aa similarity between BHG-GCoVs and GCoVs from *Charadriiformes* (AvXc-GCoV, AvNp-GCoV, AvAi-GCoV, AvCt-GCoV, AvCa-GCoV, AvTt-GCoV, AvLl-GCoV, excluding the common gull GCoV) ranges from 93.93% to 99.83%. The similarity with *Anseriformes* (AvAc-GCoV, AvAa-GCoV, AvAp-GCoV, AvAz-GCoV, AvMp-GCoV, AvMs-GCoV, Shelduck_CoV, Duck_GCoV 2714) is only from 47.19% to 69.26%, while the similarity with the common gull GCoV is 70.12%.

### 3.4. Phylogenetic Analysis of BHG-GCoVs

Based on the whole-genome nucleotide sequences, a phylogenetic tree was constructed using the maximum likelihood method with GTR+G+I as the best substitution model and 1,000 bootstrap replicates with other CoVs. The phylogenetic tree indicated that the two BHG-GCoVs clustered within the species *Gammacoronavirus anatis* in the subgenus *Igacovirus* of the genus *Gammacoronavirus*, represented by Duck CoV_2714 (NC_048214). Additionally, the BHG-GCoVs formed a sub-branch with AvXc-GCoV and AvNp-GCoV, which were recently detected in Shanghai, China (Figure 4).

### 3.5. Phylogenetic Evolution of the Conserved Domains of BHG-GCoVs

Phylogenetic trees were constructed using the aa sequences of five conserved domains (3CLpro, NiRAN, RdRp, ZBD, HEL1) in non-structural proteins, as well as the S protein, for comparison with other GCoVs. The results showed that BHG-GCoVs clustered in the same sub-branch with AvXc-GCoV and AvNp-GCoV within the 3CLpro, ZBD, and HEL1 domains. In the NiRAN and RdRp domains, BHG-GCoVs were grouped in the same sub-branch with common gull GCoV. Notably, regarding the S protein, BHG-GCoVs were grouped in the same sub-branch as AvAi-GCoV (Figure 5).

### 3.6. Recombination Analysis of BHG-GCoVs

We found that two strains of BHG-GCoVs changed their positions in the phylogenetic trees based on different domains, indicating that recombination events may occur frequently during virus evolution. Screening for potential recombination events was conducted on the whole-genome sequences of the two BHG-GCoVs in this study. The most significant recombination events in DLU1 and DLU2 were identified using the GENECONV (*p*-value: 8.66 × 10^−243^) and RDP (*p*-value: 2.035 × 10^−251^) methods provided by the RDP v4.100 program. These recombination events, characterized by strong *p*-values, were further confirmed using the SimPlot v3.5.1 program. The results indicated that BHG-GCoVs are likely recombinant strains derived from common gull GCoV and AvAi-GCoV, with breakpoints at genomic positions 20,156 bp and 24,432 bp located near the start and end regions of the S protein (Figure 6).

### 3.7. Evolutionary Rate and tMRCA

The average evolutionary rate of CoVs in the *RdRp* gene was estimated to be 1.58 × 10^−4^ subs/site/year (7.60 × 10^−5^–2.37 × 10^−4^, 95% HPD) using Bayesian Skyline under a relaxed molecular clock model with an uncorrelated lognormal distribution. Molecular clock analysis of the *RdRp* gene revealed that the most recent common ancestor (tMRCA) of avian γ-CoV and beluga whale γ-CoV diverged around 3934 B.C. (95% highest posterior density (HPD): 9437 B.C.–542 B.C.). The tMRCA differentiation time for the subgenus *Brangacovirus* and *Igacovirus* is approximately 246 B.C. (95% HPD: 2244 B.C.–887 A.D.). For the subgenus *Igacovirus*, the tMRCA is estimated to be around 788 A.D. (95% HPD: 153 B.C–1323 A.D.). The tMRCA for AvXc-GCoV is approximately 1984 A.D. (95% HPD: 1950 A.D.–2006 A.D.), while that for BHG-GCoVs is about 1994 A.D. (95% HPD: 1966 A.D.–2011 A.D.) (Figure 7).

## 4. Discussion

Since the CoVs that can infect humans are primarily α-CoV and β-CoV, current surveillance efforts focus mainly on these two groups and their host animals, such as bats and rodents [32,43,44,45]. However, on a global scale, birds also serve as important hosts for some zoonotic viruses. The wide variety and abundance of bird species, their ability to fly long distances, and their close relationships with humans and domestic animals significantly contribute to the transmission of pathogens. Thus, it is essential to monitor pathogens carried by birds. Although there have been no reports of CoVs directly infecting humans from poultry or wild birds, a case of human infection by porcine deltacoronavirus (PDCoV) in Haiti, reported in 2021, suggests that δ-CoVs and γ-CoVs, which primarily infect poultry and birds, may also have the potential for cross-species transmission to livestock and subsequently infect humans [46].

Black-headed gulls are natural hosts of *δ-CoV* and *γ-CoV*. The present study investigated CoVs in black-headed gulls, which appear to have a higher prevalence in this region compared to others. For example, the prevalence of BHG-CoVs was found to be 8.20% (5/61) in the Bering Strait, while in Poland, it was 4.53% (35/773). In this study, the prevalence of γ-CoVs was higher than that of δ-CoVs, consistent with findings from other research [23,29].

We successfully obtained CoV-positive samples and selected two BHG-GCoVs derived from black-headed gulls for whole-genome sequencing, resulting in the complete genomes of these two BHG-GCoVs. To date, the whole genome of BHG-GCoV has not been reported, DLU1 and DLU2 are the first two complete genomes of BHG-GCoVs that have been sequenced.

Based on the BLAST of the whole genomes of BHG-GCoVs available on GenBank, we found that BHG-GCoVs belong to the subgenus *Igacovirus* within the genus *Gammacoronavirus*. The subgenus *Igacovirus* is divided into three species, namely *Gammacoronavirus anatis*, *Gammacoronavirus galli*, and *Gammacoronavirus pulli* (https://ictv.global/taxonomy, accessed on 19 December 2024). The latter two species both belong to infectious bronchitis virus (IBV), while BHG-GCoVs show a closer similarity to the species *Gammacoronavirus anatis*. Previously, the number of samples with whole-genome sequences or relatively long sequences from the species *Gammacoronavirus anatis* has been limited, being primarily sourced from Poland, Australia, and China [30,47,48,49,50]. Recently, some complete genome sequences belonging to the species *Gammacoronavirus anatis* from 13 different hosts of 3 families of 2 orders were revealed in China [51], which provide an important reference basis for research on the cross-species transmission of BHG-GCoVs. Phylogenetic analysis of the genomes indicated that BHG-GCoVs belong to the species *Gammacoronavirus anatis* of the subgenus *Igacovirus*. BHG-GCoVs clustered into the same sub-branch as AvXc-GCoV and AvNp-GCoV, suggesting a close genetic relationship. However, their genetic relationship with the representative strain of Duck_GCoV 2714 is more distant.

By comparing the genome and five domain sequences of BHG-GCoVs with other GCoVs, we found that the genomic length of BHG-GCoVs is similar to those of other GCoVs from *Charadriiformes*, averaging about 27,300 bp, whereas GCoVs from *Anseriformes* tend to be longer than 28,000 bp (Appendix A). Moreover, we observed that GCoVs carried by *Charadriformes* are more similar to BHG-GCoVs than those carried by *Anseriformes*. Thus, it appears that GCoVs exhibit higher similarity within hosts from the same order. Similarly, the similarity of the S gene of BHG-GCoVs to that of other GCoVs from *Charadriiformes* is significantly greater than that of GCoVs from *Anseriformes*. Additionally, it is noteworthy that although both the common gull and the black-headed gull belong to the family *Laridae*, there is a substantial difference in the S protein of the GCoVs they carry, with similarities ranging from 54.51% to 54.59%. Whether this variation is due to species differences or regional differences requires further investigation.

The S protein of CoV plays a crucial role in binding to receptors on host cells and facilitating the fusion of the viral membrane with the host cell membrane. During the maturation of CoV, host proteases recognize and cleave the extracellular region of the S protein into two subunits, S1 and S2 [52]. The S1 subunit is primarily responsible for binding to cell receptors for viral adsorption, while the S2 subunit mainly mediates membrane fusion with the host cell, enabling the internalization of the virus [53]. Further analysis of the S protein of BHG-GCoVs revealed that most mutations occur in the S1 subunit, whereas the S2 subunit is more conserved [30].

Interestingly, the positions of the two strains of BHG-GCoVs changed in the phylogenetic tree when analyzed based on different genes, particularly the S protein. This led us to hypothesize that a recombination event may have occurred. A subsequent analysis confirmed that DLU1 and DLU2 are recombinant strains derived from common gull GCoV and AvAi-GCoV. Additionally, estimates based on temporal divergence indicated that the tMRCA of GCoVs from *Scolopacidae* predates that of BHG-GCoVs. Therefore, it is likely that BHG-GCoVs evolved from GCoVs found in from *Scolopacidae* and common gull GCoV.

Health threats posed by CoVs are persistent and long-term. γ-CoVs exhibit greater genetic and host diversity, yet our understanding of them remains limited. Continuous surveillance of CoVs in birds and poultry is essential to prevent the emergence of future infectious diseases.

## 5. Conclusions

In this study, both GCoVs and DCoVs were detected in the fecal samples of black-headed gulls, with a higher prevalence of BHG-GCoVs compared to BHG-DCoVs. Two strains of BHG-GCoVs had their complete genomes amplified and characterized. Phylogenetic analysis of these complete genomes showed that BHG-GCoVs belong to the species *Gammacoronavirus anatis* within the subgenus *Igacovirus* of the genus *Gammacoronavirus*. Recombination analysis indicated that BHG-GCoVs are recombinant strains derived from common gull GCoV and AvAi-GCoV, suggesting that cross-species transmission events may frequently occur during the viral evolution.

## Figures and Tables

**Figure 1 microorganisms-13-00874-f001:**
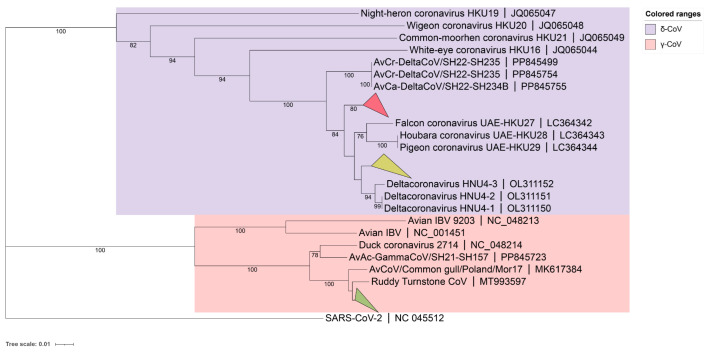
A phylogenetic tree was constructed using the neighbor-joining method based on the partial RdRp nucleotide sequences. The red collapsed branch represents the 10 strains of BHG-DCoVs detected in this study. The yellow collapsed branch represents the 8 strains of BHG-DCoVs detected in this study. The green collapsed branch represents the 41 strains of BHG-GCoVs detected in this study.

**Figure 2 microorganisms-13-00874-f002:**
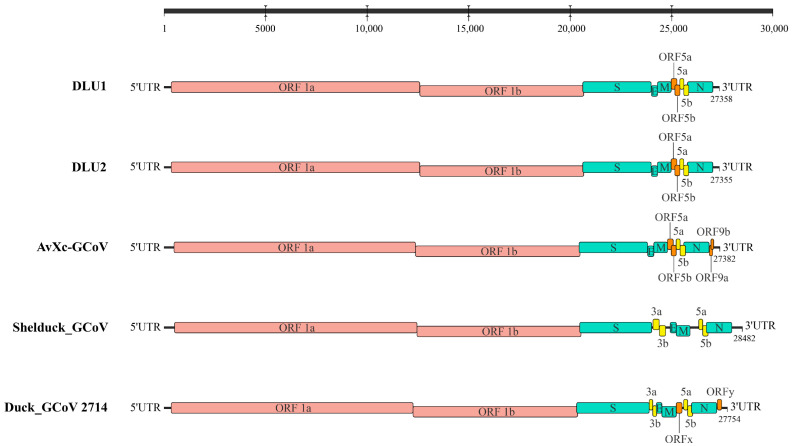
Genome structures of five GCoVs. Red represents ORF1a/b, green represents structural proteins, yellow represents accessory proteins, and orange represents putative proteins. The GenBank accession numbers corresponding to AvXc-GCoV, Shelduck_GCoV, and Duck_GCoV 2714 are PP845453, MK204411, and NC_048214, respectively.

**Figure 3 microorganisms-13-00874-f003:**
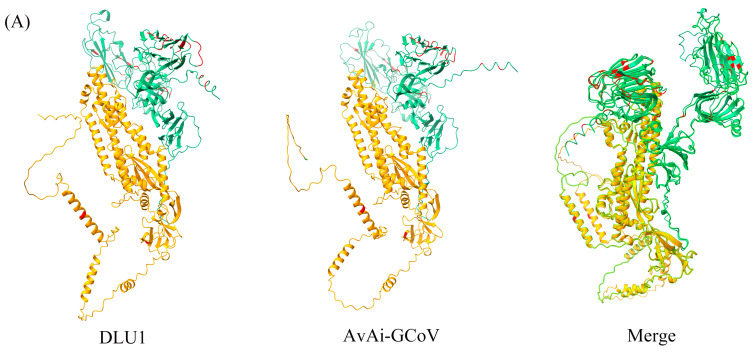
(**A**) Tertiary structure of the S protein of DLU1 and AvAi-GCoV. Green represents the S1 subunit, yellow represents the S2 subunit, and red represents the amino acid differences in the S proteins of DLU1 and AvAi-GCoV. (**B**) Multiple alignment of the amino acids of the S protein of DLU1, AvAi-GCoV, and common gull GCoV. The red box represents the amino acid differences between DLU1 and AvAi-GCoV. There is an interval of 20 nucleotides between two adjacent “*”.

**Figure 4 microorganisms-13-00874-f004:**
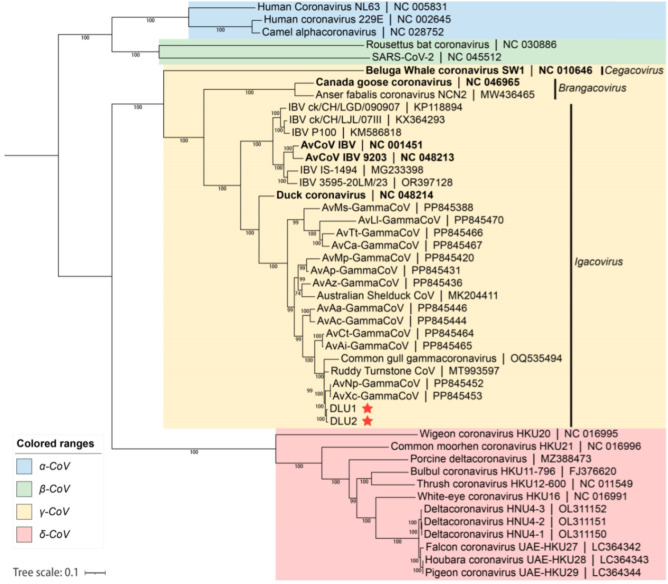
The blue background represents the genus *Alphacoronavirus*, the green background represents the genus *Betacoronavirus*, the yellow background represents the genus *Gammacoronavirus*, and the red background represents the genus *Deltacoronavirus*. The red stars indicate DLU1 and DLU2 detected in this study. The bold fonts represent the representative strains of the genus *Gammacoronavirus*.

**Figure 5 microorganisms-13-00874-f005:**
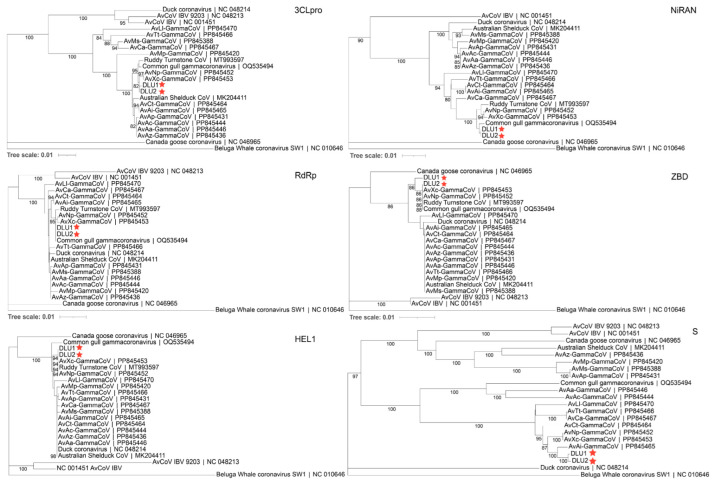
The phylogenetic tree was constructed using the neighbor-joining method based on the amino acid sequences of 3CLpro, NiRAN, RdRp, ZBD, HEL1, and S. The red stars represent DLU1 and DLU2 detected in this study.

**Figure 6 microorganisms-13-00874-f006:**
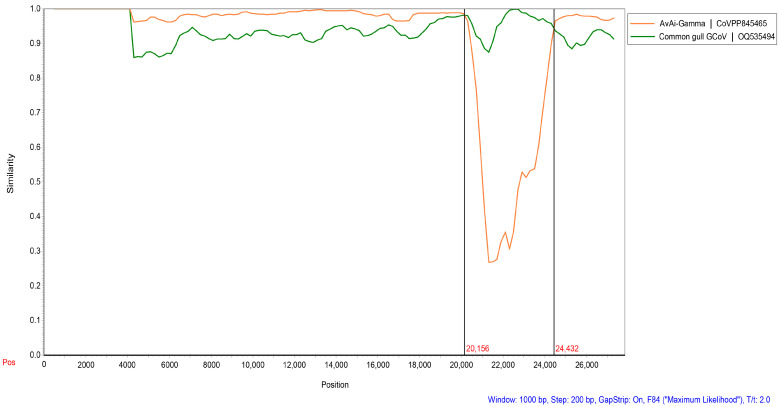
Detection of potential recombination events of BHG-GCoVs. The similarity plot method in SimPlot v3.5.1 adopted the F84 distance model, with a window size of 1000 bp and a step size of 200 bp. Common gull GCoV (OQ535494) is a non-whole genome.

**Figure 7 microorganisms-13-00874-f007:**
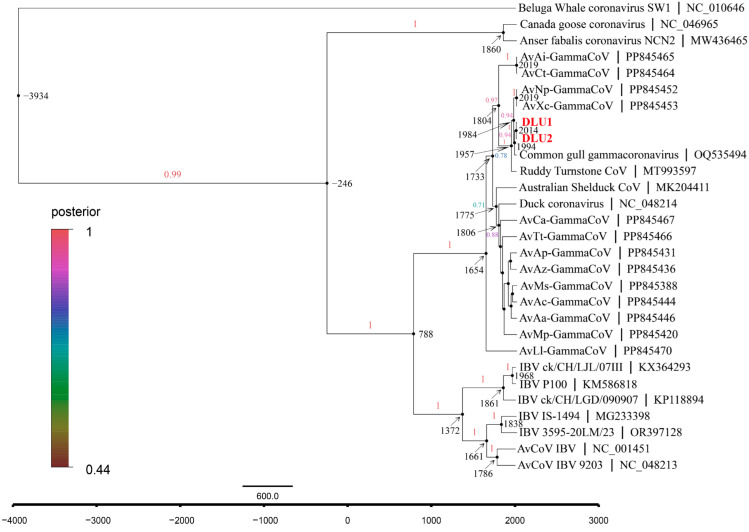
The maximum clade credibility (MCC) tree with divergence times based on the *RdRp* gene. The red fonts represent DLU1 and DLU2 detected in this study.

**Table 2 microorganisms-13-00874-t002:** Detection results of coronaviruses by RT-PCR in black-headed gulls.

Sampling Site	γ-CoV		δ-CoV	
	Prevalence	95%CI	Prevalence	95%CI
Cuihu Park	4.44% (10/225)	1.73–7.16%	1.78% (4/225)	0.04–3.52%
Daguanlou Park	15.00% (12/80)	7.00–23.00%	12.50% (10/80)	5.09–19.91%
Haigeng Dam	9.31% (19/204)	5.29–13.34%	1.96% (4/204)	0.04–3.88%
Total	8.06% (41/509)	5.68–10.43%	3.54% (18/509)	1.90–5.10%

## Data Availability

All the sequences in this manuscript can be obtained from the NCBI database (https://www.ncbi.nlm.nih.gov, accessed on 8 January 2025).

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
