# Peer review of "Detection of Coronaviruses and Genomic Characterization of Gammacoronaviruses from Overwintering Black-Headed Gulls (*Chroicocephalus ridibundus*) in Yunnan Province, China"

_microorganisms, 2025, doi:10.3390/microorganisms13040874_

Round 1
Reviewer 1 Report
Comments and Suggestions for Authors
- I suggest to use the same definition and rate indicator regarding length of viral genomes - f.e. bp and kbp against nt or kb used in abstract and intro parts.
- primer sets and amplification parameters on the indication stage would be better presented as a tables.
- fig. 1 and other phylogenetic trees - I suggest to add more details in NJ algorithm parameters. In the signs of compared known sequences is better to add GenBank numbers in similar way f.e. letters and numbers together or after particular isolate name.
- fig. 3 protein subunits, marked with different colours is better to indicate with sings on the pictures
- fig. 5 the range markers are to small to be readable, it is better to enlarge this fig.
- I suggest to add separate conclusion section and increase the number of last 5 y.o. references
Author Response
- I suggest to use the same definition and rate indicator regarding length of viral genomes - f.e. bp and kbp against nt or kb used in abstract and intro parts.
Response 1: Thanks. The definition of the lengths of all viral genomes have been modified to bp or kbp. Please see line 23, 232, 346, 410, 411: Changed nt to bp .Line 53: Changed kb to kbp.
- primer sets and amplification parameters on the indication stage would be better presented as a tables.
Response 2: Thanks. The primer sets and amplification parameters have been revised into a tabular format. Please see the Table 1.
- 1 and other phylogenetic trees - I suggest to add more details in NJ algorithm parameters. In the signs of compared known sequences is better to add GenBank numbers in similar way f.e. letters and numbers together or after particular isolate name.
Response 3: Thanks. Detailed parameters have been added to all the phylogenetic trees constructed using the neighbor-joining algorithm. Please see line 162-165. For all the sequences used for comparison in the phylogenetic trees, the GenBank numbers have all been modified to be placed behind the isolation names. Please see the Fig. 1, 4, 5, 6 and 7.
- 3 protein subunits, marked with different colours is better to indicate with sings on the pictures
Response 4: Thanks. The S1 and S2 subunits have been marked in different colors in Fig 3(B).Please see the Fig 3(B).
- 5 the range markers are to small to be readable, it is better to enlarge this fig.
Response 5: Thanks. The Fig 5 has been enlarged. Please see the Fig 5.
- I suggest to add separate conclusion section and increase the number of last 5 y.o. references
Response 6: Thanks. A separate conclusion section has been added, and seven references from the most recent five years have been included. Please see the conclusion section (line440-448) and the reference section (ref. 1, 16, 26, 27, 42, 46,52).
Reviewer 2 Report
Comments and Suggestions for Authors
In the context of investigations into faecal samples from 509 black-headed gulls (BHG) from Kunming city, Yunnan, China, delta and gamma coronaviruses were detected. The complete genomes of two GCoVs were obtained and analysed. Phylogenetic analyses showed a closer relationship of these viruses to GCoVs of other Charadriiformes compared to GCoVs from Anseriformes.
The data obtained complete the view of avian CoV; in particular, knowledge about the breadth of CoV occurrence is limited in wild birds. The analyses are largely meaningful and technically sound. It remains to be asked why structural analyses were carried out, since the data obtained remain somewhat incoherent in the manuscript.
Some ambiguities remain, which can partly be attributed to the poor quality of the language. Linguistic editing by a native speaker is highly recommended.
17: “…total of 59 Covs…”; misleading: You have 59 CoV-PCR positive fecal samples, not 59 coronaviruses.
30: “…five conserved domains…” Here and elsewhere a reference is required as to what nature these domains have; you could say that sequences of these create a kind of sketched profile allowing a gross classification within the CoV taxonomy.
33: “… relatively high…relatively low…” Here and elsewhere pls use more exact definitions/descriptions. No one knows what relatively high etc. means. Pls rephrase.
36: “recombination events” Pls indicate which viruses have been implicated as donors and acceptors here.
140-145: Obviously, the authors have used here a kind of tiling PCR to amplify the full length genome. The methodology employed should be described in more detail. A list of primers can be added to the supplement.
227-8: Pls check the numbers. 37355 is probably wrong meaning 27355 instead?
392: You mean the latter two?
427: Not quite; see IBV: There is an enormous variability in the IBV S gene although all isolates come from a single species (chicken) only?
Comments on the Quality of English Language
Included in the above statement.
Author Response
In the context of investigations into faecal samples from 509 black-headed gulls (BHG) from Kunming city, Yunnan, China, delta and gamma coronaviruses were detected. The complete genomes of two GCoVs were obtained and analysed. Phylogenetic analyses showed a closer relationship of these viruses to GCoVs of other Charadriiformes compared to GCoVs from Anseriformes.
The data obtained complete the view of avian CoV; in particular, knowledge about the breadth of CoV occurrence is limited in wild birds. The analyses are largely meaningful and technically sound. It remains to be asked why structural analyses were carried out, since the data obtained remain somewhat incoherent in the manuscript.
Some ambiguities remain, which can partly be attributed to the poor quality of the language. Linguistic editing by a native speaker is highly recommended.
- 17: “…total of 59 Covs…”; misleading: You have 59 CoV-PCR positive fecal samples, not 59 coronaviruses.
Response 1: Thank you for your professional comments. The revisions have been made. Please see the line 17 and 193.
- 30: “…five conserved domains…” Here and elsewhere a reference is required as to what nature these domains have; you could say that sequences of these create a kind of sketched profile allowing a gross classification within the CoV taxonomy.
Response 2: Thanks. It has been added. Please see the line 64-65.
- 33: “… relatively high…relatively low…” Here and elsewhere pls use more exact definitions/descriptions. No one knows what relatively high etc. means. Pls rephrase.
Response 3: Thanks. The revisions have been made. Please see the line 28, 33-36,208,216,268-270,276-278,282-287,405,417,427.
- 36: “recombination events” Pls indicate which viruses have been implicated as donors and acceptors here.
Response 4: Thanks. It has been added. Please see the line 36-39.
- 140-145: Obviously, the authors have used here a kind of tiling PCR to amplify the full length genome. The methodology employed should be described in more detail. A list of primers can be added to the supplement.
Response 5: Thanks. The primers used for amplifying the whole genome of BHG-GCoV in this study have been added to Table S5. Please see the line 150,454 and table S5.
- 227-8: Pls check the numbers. 37355 is probably wrong meaning 27355 instead?
Response 6: Thank you for your meticulous comments. The revisions have been made. Please see the line 23 and 232.
- 392: You mean the latter two?
Response 7: Thank you for your meticulous comments. The revisions have been made. Please see the line 394.
- 427: Not quite; see IBV: There is an enormous variability in the IBV S gene although all isolates come from a single species (chicken) only?
Response 8: Thank you for your professional comments. We have deleted this sentence.
